# Deletion of Diterpenoid Biosynthetic Genes *CYP76M7* and *CYP76M8* Induces Cell Death and Enhances Bacterial Blight Resistance in *Indica* Rice *‘9311’*

**DOI:** 10.3390/ijms23137234

**Published:** 2022-06-29

**Authors:** Min Jiang, Ning Yu, Yingxin Zhang, Lin Liu, Zhi Li, Chen Wang, Shihua Cheng, Liyong Cao, Qunen Liu

**Affiliations:** 1State Key Laboratory of Rice Biology, China National Rice Research Institute, Hangzhou 311400, China; jiang_min1997@163.com (M.J.); yuning02@163.com (N.Y.); zhangyingxin@caas.cn (Y.Z.); changdaliulin@sina.com (L.L.); lizhihpp@163.com (Z.L.); 984810137wc@gmail.com (C.W.); chengshihua@caas.cn (S.C.); 2Key Laboratory for Zhejiang Super Rice Research, China National Rice Research Institute, Hangzhou 311400, China; 3Guangdong Key Laboratory of New Technology in Rice Breeding, Rice Research Institute of Guangdong Academy of Agricultural Sciences, Guangzhou 510640, China; 4Northern Center for China National Rice Research Institute, China National Rice Research Institute, Hangzhou 311400, China

**Keywords:** lesion mimic mutants (LMMs), diterpenoid biosynthetic genes, disease resistance, lignin biosynthesis, rice (*Oryza sativa* L.)

## Abstract

Lesion mimic mutants (LMMs) are ideal materials for studying cell death and resistance mechanisms. Here, we identified and mapped a novel rice LMM, *g380*. The *g380* exhibits a spontaneous hypersensitive response-like cell death phenotype accompanied by excessive accumulation of reactive oxygen species (ROS) and upregulated expression of pathogenesis-related genes, as well as enhanced resistance to *Xanthomonas oryzae* pv. *oryzae* (*Xoo*). Using a map-based cloning strategy, a 184,916 bp deletion on chromosome 2 that overlaps with the diterpenoid biosynthetic gene cluster was identified in *g380*. Accordingly, the content of diterpenoids decreased in *g380*. In addition, lignin, one of the physical lines of plant defense, was increased in *g380*. RNA-seq analysis showed 590 significantly differentially expressed genes (DEG) between the wild-type *9311* and *g380*, 585 of which were upregulated in *g380*. Upregulated genes in *g380* were mainly enriched in the monolignol biosynthesis branches of the phenylpropanoid biosynthesis pathway, the plant–pathogen interaction pathway and the phytoalexin-specialized diterpenoid biosynthesis pathway. Taken together, our results indicate that the diterpenoid biosynthetic gene cluster on chromosome 2 is involved in immune reprogramming, which in turn regulates cell death in rice.

## 1. Introduction

Plants that spontaneously form necrotic spots on their leaves or stems are called lesion mimic mutants (LMMs) or spotted leaf mutants (SPLs). Due to the phenotype of necrotic spots and the generally concomitant changes in disease resistance, LMMs are ideal materials for deciphering the mechanisms of plant cell death and defense response from the perspective of forwarding genetics. Starting in the early 1980s, LMMs have been found in Arabidopsis, rice, maize, potato, and barley [1]. Studies of these LMMs indicate that the underlying mechanisms of cell death and defense response are complicated. Plant hormones, phytoalexins, and reactive oxygen species (ROS) have nonnegligible roles in plant cell death and defense response. In Arabidopsis mutant *atg2-2*, spontaneous cell death, early senescence, and disease resistance required the salicylic acid (SA) pathway [2]. In rice LMM *lrd6-6*, the content of antimicrobial metabolites in leaves, such as diterpenoid phytoalexins and lignin, was significantly higher than that of the wild type, and the resistance to rice blast was enhanced [3]. Defense responses and ROS are induced by the functional inactivation of OsUAP1 in the LMM *spl29* [4]. At least 40 LMM genes have been isolated in rice so far.

Phytoalexins are small molecular compounds that can increase plant resistance and can be induced when plants are under external stress. Flavonoids and diterpenoids are the main phytoalexins in rice, and the primary diterpenoids include momilatones, phytocassanes, oryzalexins, and oryzalides. Phytocassanes are essential in resistance to rice blast and sheath blight disease [5]. The biosynthetic genes for phytocassanes are clustered on chromosome 2 (*c2BGC*) in rice [6]. This diterpenoid biosynthetic gene cluster contains one type II terpene synthase gene (*OsCPS2*), three type I terpene synthases genes (*OsKSL5-7*), and six CYP450 family genes (*CYP76M5-8*, *CYP71Z6&7*) [6]. *OsCPS2* is a phytoalexin-specific *ent*-copalyl diphosphate (*ent*-CPP) synthase that converts the common substrate geranylgeranyl pyrophosphate (GGPP) into *ent*-CPP [7]. *OsKSL7* encodes *ent*-cassa-12,15-diene synthase, which catalyzes the formation of *ent*-cassa-12,15-diene from *ent*-CPP, an intermediate product of phytocassane biosynthesis [8]. *OsKSL5&6* in *indica* rice both encode *ent*-isokaurene synthase, an intermediate product of oryzalides biosynthesis [9]. *CYP76M5-8* and *CYP71Z6&7* encode hydroxylases that modify the intermediate products of diterpenoid phytoalexin biosynthesis [10]. Another major diterpenoid biosynthetic gene cluster on the rice genome is located on chromosome 4 (*c4BGC*) and mainly controls momilactone biosynthesis [11]; *c4BGC* contains *OsCPS4*, *OsKSL4*, *CYP99A2*, and *CYP99A3* and a dehydrogenase gene *OsMAS* [11]. Simultaneous knockout of *CYP76M7* and *CYP76M8* on *c2BGC* resulted in a lesion mimic phenotype in rice [12]. Moreover, after knocking out the entire diterpenoid *c2BGC*, the lesion phenotype persisted; however, in the absence of both gene clusters (*c2BGC* and *c4BGC*), the lesion phenotype disappeared [13].

Lignin is a component of the plant secondary cell wall, enhancing mechanical strength and resistance to pathogens [14]. Lignin can be induced by biotic or abiotic stresses, such as wounding and pathogen infection [15]. The structure of lignin is very complex, and it is mainly composed of one or more of the three monolignols *p*-hydroxyphenyl lignin, guaiacyl lignin, and syringyl lignin [16]. The biosynthesis of lignin occurs through the phenylpropanoid biosynthesis pathway, which also includes flavonoid biosynthesis. Ten key enzymes are involved in biosynthesis from phenylalanine to monolignols: phenylalanine ammonia lyase (PAL), cinnamate 4-hydroxylase (C4H), 4-coumarate: CoA ligase (4CL), hydroxycinnamoyl CoA: shikimate hydroxycinnamoyl transferase (HCT), *p*-coumaroyl shikimate 3′-hydroxylase (C3′H), caffeoyl CoA O-methyltransferase (CCoAOMT), (hydroxy)cinnamoyl CoA reductase (CCR), ferulate (coniferaldehyde) 5-hydroxylase (F5H), caffeic acid (5-hydroxy-coniferaldehyde) O-methyltransferase (COMT), and (hydroxy)cinnamyl alcohol dehydrogenase (CAD) [15]. Usually, the genes encoding these enzymes have multiple homologs in rice.

In this study, we isolated a new LMM, *g380*, in the background of *indica* rice cultivar ‘*9311′*. Reddish-brown lesions appeared in lower leaves at around 40 days after sowing (DAS). Map-based cloning localized the *g380* mutation to a 184,916 bp deletion on the long arm of chromosome 2. The deletion contains most genes of *c2BGC*. High-level ROS accumulation and the ROS scavenging enzyme system of *g380* were disturbed. The defense response was activated prematurely and *g380* was more resistant to bacterial blight. RNA-seq results suggested that the phenylpropanoid biosynthesis pathway was upregulated in *g380.* The upregulated genes in *g380* enriched in this pathway are mainly distributed in the monolignol biosynthesis branches. Lignin content in the leaves of *g380* was also higher than that of *9311*. Pathogenesis-related genes were upregulated supporting spontaneous defense response activation in *g380*. Our results suggested that the deletion on chromosome 2 induces spontaneous cell death and enhances resistance to bacterial blight in rice.

## 2. Results

### 2.1. Characterization of the g380 Mutant

An LMM was isolated from a *^60^Co* γ-ray treated rice mutant library (*Oryza sativa* L. ssp. *Indica* cv. ‘*9311′*) and named *g380*. There was no difference between *g380* and wild-type *9311* plants before the tillering stage, but reddish-brown lesions began to appear on the lower leaves of *g380* at the tillering stage (about 40 DAS). Over time the lesions became larger and more oval and spread to all leaves (Figure 1a–c). Compared with *9311*, *g380* plant height was significantly shorter, the spikelet fertility was lower, and the primary branch number per panicle was higher (Figure 1d–f). No significant differences were observed for the number of panicles per plant, panicle length, or thousand-grain weight of *9311* and *g380*.

Since leaf chlorophyll directly affects photosynthesis and, therefore, crop growth rate, we determined the chlorophyll content. Chlorophyll a, chlorophyll b, and total chlorophyll were significantly reduced in *g380* compared to *9311* (Figure 1g–i). The results suggested that *g380* plants have poorer photosynthesis and a lower crop growth rate.

### 2.2. Fine Mapping of the g380 Locus

To locate *g380*, we adopted a map-based cloning strategy. First, we found that *g380* was located on the long arm of chromosome 2, linked to the markers RM341 and RM3850 through linkage marker screening. Next, we analyzed 104 phenotypic F_2_ plants and narrowed the range to 2470 kb between C2-9 and C2-10 (Figure 2a). Then, we designed new InDel primers between these two markers based on the polymorphisms of *9311* and Nipponbare genome sequences to continue to analyze 2799 plants with the lesion phenotype in F_2_ and narrowed the location range to 248 kb between L26 and L39 (Figure 2b,c).

Using chromosome walking, we found that the amplicon with primer L10 could not be formed from the recessive *g380* phenotypic plants, which indicates a fragment deletion at the L10 site in *g380* (Figure 2e). To confirm the deletion range, ten new primers surrounding L10 (5 kb upstream and 5 kb downstream) were used to amplify *9311* and *g380*. All ten primers failed to obtain amplicons from *g380* but succeeded with *9311* (data not shown), suggesting a large fragment deletion in *g380*. Additional primers were used to expand the analysis range, leading to identification of the left end of the deletion site between L27 and L28 and the right end of the deletion site between L38 and L39 (Figure 2f). Finally, the deletion site was narrowed to the ranges between F13 and F4 (left), R11 and R10 (right) (Figure 2g). Then, we performed PCR with the forward primer F13 and reverse primer R10, and an amplicon of about 3 kb in size was obtained from *g380*. The amplicon was sequenced and aligned with the *9311* genome sequence (http://plants.ensembl.org/Oryza_indica/Location/Genome, accessed on 21 June 2022) (Figure 2h).

The result showed a 184,916 bp (location: 2:23,361,216–23,546,131) fragment deletion in *g380* relative to the wild-type *9311* (Figure 2c,h). The deletion fragment was aligned to a 224 kb (location: 21,603,461–21,827,464) sequence of the Nipponbare reference genome containing eight annotated genes (Figure 2d). Six of the eight genes belong to the diterpenoid biosynthesis gene cluster on chromosome 2 (*c2BGC*), which is involved in the biosynthesis of various phytoalexins and functions directly in resistance to rice blast and sheath blight disease [5]. *OsKSL7* encodes an *ent*-cassa-12,15-diene synthase; *CYP76M5*, *CYP76M8*, *CYP76M7*, *CYP71Z6*, and *CYP71Z7* encode putative proteins in the CYP450 family. Of the two remaining genes, *CPT1* encodes an NPH3 domain-containing protein, which is a key signal transduction component of higher plant phototropism [17], and *LOC_Os02g36000* encodes a putative zinc finger protein. Furthermore, Ye et al. reported that co-knockdown of *CYP76M7* and *CYP76M8* resulted in a lesion-mimic phenotype in rice [12], and Li et al. found that *CYP76M7* and *CYP76M8* double knockout mutant *cyp76m7/m8* and *c2BGC* deletion mutant *Δc2bgc* also have lesion-mimic phenotype, while single knockout mutants of *CYP76M7* or *CYP76M8* do not show lesion-mimic phenotype [13]. These findings revealed that lesion-mimic phenotype of the *g380* mutant comes from the deletion of *CYP76M7* and *CYP76M8* on *c2BGC*.

### 2.3. ROS Accumulation in g380

Reactive oxygen species (ROS) play a vital role in signaling pathways that regulate defense responses in plants. The high accumulation of ROS can trigger redox homeostasis disturbance, which can lead to cell death and, consequently, to limiting biomass and yield production [18]. To ascertain whether the appearance of *g380* lesions is related to ROS accumulation, we performed diaminobenzidine (DAB) staining and detected H_2_O_2_ content. There was an obvious accumulation of H_2_O_2_ around the lesions of *g380* leaves (Figure 3a), and total H_2_O_2_ in *g380* leaves was significantly higher than that of *9311* (Figure 3b).

To further investigate the biochemical mechanisms of ROS accumulation, we measured the activities of catalase (CAT), peroxidase (POD), and total superoxide dismutase (T-SOD). The activity of T-SOD was significantly lower in *g380* than in *9311*, while CAT activity was significantly increased in *g380* compared to *9311*, the POD activity had no significant difference between *9311* and *g380* (Figure 3c–e). Malondialdehyde (MDA) is produced and accumulated in the cell due to membrane lipid peroxidation. To determine the extent of membrane damage, we measured the content of MDA. MDA content was also significantly increased in *g380* (Figure 3f). These results suggested that high-level ROS accumulated in *g380* and the ROS scavenging enzyme system of *g380* was disturbed.

To explore the order of appearance between ROS accumulation and lesion formation, we performed DAB staining and measured the ROS indexes of *g380* and *9311* of 30 DAS when the lesion did not appear in *g380*. The DAB staining result did not show obvious difference between *9311* and *g380*. However, CAT and POD activities andH_2_O_2_ content increased, T-SOD activity decreased in *g380*, while MDA content did not show significant difference (Appendix A). ROS accumulation appears to precede lesion appearance in *g380*.

### 2.4. Activated Defense Response and Increased Resistance to Bacterial Blight in g380

The upregulated expression of some pathogenesis-related (*PR*) genes, such as *OsPR1a*, *OsPR1b*, and *OsPR10,* is a marker of the defense response activation and can be found in many LMMs [2,3,4]. We measured the transcript levels of *OsPR1a*, *OsPR1b*, and *OsPR10* by qRT-PCR. The expression levels of these genes were all significantly increased in *g380* (Figure 4a–c), suggesting that the defense response of *g380* is activated.

To examine whether the activated defense response would lead to enhanced disease resistance, three strains of *Xanthomonas oryzae* pv. *oryzae* (*Xoo*) were tested (Figure 4d,e). The lesion lengths on *9311* leaves were significantly longer than *g380* after inoculation, which indicated that *g380* exhibited enhanced resistance to *Xoo*.

### 2.5. RNA-Seq Data Output and DEG Analysis

After RNA-seq, we obtained 331,699,466 and 312,292,012 raw reads in total from *9311* and *g380* samples, respectively. Through screening, we obtained about 309,919,072 (93.43%) and 293,284,164 (93.91%) clean reads from each kind of sample (Appendix A). Among those clean reads, approximately 235,273,004 (75.91%) from *9311*, and 225,019,538 (76.72%) from *g380* were mapped to the reference genome. In *9311* and *g380*, there were 17,829 and 18,582 expressed genes with FPKM values greater than 1 in at least one replicate, respectively. A total of 17,485 genes were expressed in both *9311* and *g380*, 344 genes were expressed only in *9311*, and 1097 genes were expressed only in *g380* (Figure 5a).

By analyzing differentially expressed genes (DEGs) with edgeR, we obtained 591 genes that were significantly differentially expressed in *9311* and *g380*, of which five DEGs were downregulated in *g380* relative to *9311*. Three of these five downregulated genes were located in the deletion area. The remaining two were putative expressed proteins, and no more detailed annotations were available. So, the subsequent analyses were based on 585 upregulated DEGs.

We selected ten genes for qRT-PCR to validate the reliability of the DEGs. The RT-qPCR results were consistent with the obtained DEGs (Appendix A), demonstrating that these DEGs are reliable and can be used for subsequent analysis.

*PR* genes and *WRKY* genes are essential for defense responses. Thus, we analyzed the *PR* genes and *WRKY* genes in DEGs. The 20 *PR* genes, which belong to different *PR* gene families, had higher transcript levels in *g380* leaves compared with *9311* (Appendix A). The 7 *WRKY* genes showed higher expression levels in *g380* (Appendix A). Among these *WRKY* genes, *WRKY32* has the highest Log_2_FC value of 4.79.

To check the gene ontology to which the obtained DEGs belong, we performed GO enrichment analysis. A total of 378 DEGs were significantly enriched in molecular function and biological process. In the molecular function ontology, the top three terms were catalytic activity (GO:0003824), oxidoreductase activity (GO:0016491), and iron ion binding (GO:0005506), suggesting that upregulated genes mainly function in redox reactions and iron ion combinations. The top three terms for biological process were oxidation reduction (GO:0055114), carbohydrate metabolic process (GO:0005975), and amine metabolic process (GO:0009308), indicating that upregulated genes are mainly involved in redox reactions, carbohydrate metabolism, and amine metabolism (Figure 5b).

KEGG pathway enrichment analysis was performed to identify the pathways in which those DEGs participate. The result showed that most DEGs that had KEGG annotations belonged to the metabolic pathway and the biosynthesis of secondary metabolites. Furthermore, 109 DEGs were significantly enriched in five pathways: phenylpropanoid biosynthesis (dosa00940), plant–pathogen interaction (dosa04626), diterpenoid biosynthesis (dosa00904), glutathione metabolism (dosa00480), and MAPK signaling pathway (dosa04016) (Figure 5c). Most of the upregulated genes enriched in phenylpropanoid biosynthesis pathway are genes involved in lignin biosynthesis, which are essential in plant resistance (Appendix A). In the plant–pathogen interaction pathway, the upregulated genes are mainly *PR* genes (Appendix A). There are ten upregulated genes and four deleted genes enriched in diterpenoid biosynthesis. Among the ten genes enriched in glutathione metabolism, nine genes encode glutathione S-transferase and the remaining gene encodes ascorbate peroxidase. The MAPK signaling pathway enriched four *PR* genes, two chitinase genes, *OsWRKY24*, *OsCATA*, *OsMPK3*, and *Osrboh7*.

### 2.6. Decreased Diterpenoid Biosynthesis in g380 Leaves

Diterpenoid phytoalexins, such as momilactones and phytocassanes, have a vital role in rice defense [5]. In this study, *c2BGC* that mainly controls phytocassanes biosynthesis is mostly missing (Figure 2d and Figure 6a) [19]. Moreover, KEGG pathway enrichment analysis suggested that upregulated genes in *g380* were significantly enriched in diterpenoid biosynthesis (Figure 5c and Figure 6a) [19]. Since the *g380* lesion phenotype was associated with ROS defense signaling molecules (Figure 3), and the diterpenoid phytoalexin was induced by defense responses, we could not determine whether the upregulation of diterpenoid biosynthesis genes is due to the lesion phenotype-related defense response induction.

Based on this, we used qRT-PCR to detect the expression levels of two and three genes (*OsCPS2*, *CYP76M6*, and *OsCPS4*, *OsKSL4*, *CYP99A2*) present on chromosomes 2 and 4, respectively, in *9311* and *g380*, before and after *g380* lesion formation (Figure 6b–f). The expression levels of these genes in *g380* were significantly upregulated before and after lesion formation. This finding suggested that the upregulation of diterpenoid biosynthesis genes before *g380* lesion formation is due to the deletion of the majority of the diterpenoid *c2BGC*, rather than induced by the lesion phenotype. It is worth noting that the relative expression levels of these diterpenoid biosynthesis genes were significantly higher after the formation of *g380* lesions than before. This indicates that the upregulation of diterpenoid biosynthesis genes after lesion formation was mainly caused by the lesion-related defense response.

As mentioned above, *g380* has enhanced resistance to bacterial blight. We thus wondered whether the upregulated expression of the diterpenoid biosynthesis genes leads to consistent changes in diterpenoid content in *g380* leaves, and thereby enhances *g380* disease resistance. To answer this question, we measured the content of three major diterpenoids in *9311* and *g380* leaves before and after the lesions formed in *g380* (Figure 6g–i). The content of diterpenoid phytoalexins in *g380* was less than that in *9311* before and after the formation of *g380* lesions. In *g380* leaves, phytocassanes cannot be detected before and after the formation of *g380* lesions; momilactone A and oryzalexin A and C can be detected after the formation of *g380* lesions, but they also cannot be detected before lesions formed. Although lesion formation induced diterpenoid phytoalexins production in *g380* leaves, overall, *g380* had a lower diterpenoid phytoalexins content than *9311*, which would not account for the enhanced resistance after lesion formed in *g380*.

### 2.7. Increased Lignin Biosynthesis in g380

Lignin is part of the physical defense of plants against the invasion of pathogens [14]. Through KEGG pathway enrichment analysis, we found that genes in the lignin biosynthesis branch of the phenylpropanoid biosynthesis pathway were significantly up-regulated. This suggested that the lignin biosynthesis pathway has been activated in *g380* plants at the transcriptome level. To further confirm this result, we measured the expression levels of some lignin biosynthesis genes, such as *OsPAL1*, *OsPAL4*, *OsF5H1*, *OsROMT9*, *OsCCR20*, and *OsCAD2*, in *9311* and *g380* (Figure 7a–f). We also determined the lignin content in flag leaves of *9311* and *g380* plants (Figure 7g). The results showed that in the leaves of *g380* plants with the lesions, the expression of lignin biosynthesis genes and the content of lignin were significantly higher than those of *9311*.

## 3. Discussion

*CYP76M7* and *CYP76M8* are located on *c2BGC* and both encode CYP450 family proteins. It is reported that CYP76M7 protein function as *ent*-cassadiene C11α-hydroxylase, which play a role in phytocassane biosynthesis [6]. CYP76M8 protein is multifunctional, and redundant with CYP76M5–7 proteins [20]. Ye et al. reported that the simultaneous knockdown of *CYP76M7* and *CYP76M8* in the background of Nipponbare resulted in the formation of lesions on leaves of adult plants, and the accumulation of 3α-hydroxy-*ent*-cassadiene and 3α-hydroxy-*ent*-cassadien-2-one [12]. Therefore, they speculated that the substrates of *CYP76M7* and *CYP76M8* might be 3α-hydroxy-*ent*-cassadiene and 3α-hydroxy-*ent*-cassadien-2-one, and that the formation of lesions might be caused by cytotoxicity of these two phytocassane intermediates [12]. In that study, the formation of lesions was not caused by the pathogen invasion of the mutants, similar to our mutant *g380*, which can spontaneously form a lesion mimic phenotype. In addition, the deletion fragment of *g380* on chromosome 2 contains *CYP76M7* and *CYP76M8*. However, in our study, the type I terpene synthases *OsKSL7*, which is a key enzyme in the biosynthesis of phytocassane, is also missing. The biosynthesis of precursors of 3α-hydroxy-*ent*-cassadiene and 3α-hydroxy-*ent*-cassadien-2-one is blocked, yet the lesion phenotype still exists. It suggested that the formation of lesions is probably not caused by the accumulation of 3α-hydroxy-*ent*-cassadiene and 3α-hydroxy-*ent*-cassadien-2-one. This is supported by the research of Li et al. who found that *CYP76M7* and *CYP76M8* single knockout mutants, *c4BGC* knockout mutants, and *c2BGC* and *c4BGC* double knockout mutants did not show a lesion phenotype, while *CYP76M7* and *CYP76M8* double knockout mutants and *c2BGC* knockout mutants showed a lesion phenotype [13]. They speculated that the formation of lesions was related to the accumulation of *c4BGC* metabolites, which are the substrate of CYP76M7 and CYP76M8. The accumulation of these substrates leads to the appearance and gradual severity of cell death lesions. These results comprehensively identified that the genes responsible for *g380* lesion phenotype are *CYP76M7* and *CYP76M8*. Although *CYP76M7* and *CYP76M8* RNAi mutants, knockout mutants, and *g380* all showed lesion phenotype, the first two were more similar, which have smaller and denser lesions than *g380.* The reason for the difference might be the background*―* the background of *g380* is *Indica* rice *9311*, RNAi mutant has the background of Nipponbare, and knockout mutant has the background of *kitaake japonica* rice. It also shows that the simultaneous deletion of *CYP76M7* and *CYP76M8* leads to the emergence of lesion phenotype in both *japonica* and *indica* rice. In addition, we found the interesting phenomenon that *g380* cultured in greenhouses did not show a lesion phenotype. The biggest difference between our greenhouse and field culture conditions lies in the different spectral components. This shows that spectral components are vital for the generation of the lesions. Chloroplast is the main site of photosynthesis and one of the main sites of ROS production. The production of ROS in chloroplasts is closely related to light-dependent photosynthesis. In our study, ROS of *g380* cultured in the field began to accumulate before the appearance of lesions (Appendix A). We speculated that ROS accumulation may also be one of the causes of lesion formation. Apart from the aforementioned differences, the variation trends of oryzalexin content in *c2BGC* knockout mutants and *g380* were also inconsistent. In the *c2BGC* knockout mutants, the total content of oryzalexin A–F was not significantly different from that of the wild-type [13]. In our study, the total amount of oryzalexin A and C showed significant reduction. The cause of this difference remains to be studied.

Most notably, both the diterpenoid *c2BGC* deletion mutant and *g380* had enhanced bacterial blight resistance, upregulated expression of *PR* genes, and accumulated H_2_O_2_ in leaves after the formation of lesions [13]. It seems that the hypersensitive response (HR) was presented in the *c2BGC* deletion mutant and *g380*. The HR to a pathogen is one of the most efficient defense mechanisms in nature and leads to the induction of *PR* genes, ROS burst, and cell death [21]. *PR* proteins are conserved in higher plants and are also involved in signaling and stress response [22]. They are classified into 17 families, based on the similarity of protein sequence, serology relationship, and enzyme activity [23]. In our study, 20 *PR* genes belonging to nine families were up-regulated (Appendix A), and previous studies showed that many were induced after inoculation with *Magnaporthe grisea* and *Xoo*. Furthermore, overexpression was demonstrated to enhance the disease resistance for some genes [24,25,26,27]. In addition, our RNA-seq results showed that seven *WRKY* genes exhibited enhanced transcripts in *g380* (Appendix A). Previous studies have also shown that *WRKY* genes are involved in the defense against attack from pathogenic bacteria and fungi, and responses to abiotic stress, such as wounding, drought, and cold [28]. Among them, the expression level of *WRKY28* can be induced by infection with *M. grisea*, but it acts as a negative regulator of innate immunity in rice [29]. We also found that the expression levels of *PAL1* and *OsWRKY19* were increased (Figure 7, Appendix A). Studies have reported that *OsRac1* is a key defensive response regulator, and *OsPAL1* and *OsWRKY19* are regulated by the downstream transcription factor *OsRAI1* of *OsRac1* [30].

Lignin is the second most abundant plant biopolymer after cellulose and is mostly deposited in the secondary cell walls of vascular plants [14]. Besides water transport and mechanical support, lignin also aids pathogen defense by accumulating throughout the HR region as a physical barrier against infection by pathogens [31]. Our study found that DEGs involved in the phenylpropanoid biosynthesis pathway mainly participate in the biosynthesis of hydroxyphenyl lignin, guaiacyl lignin, and syringyl lignin, which are the monolignols incorporated into lignin polymers. The lignin content in *g380* flag leaves was significantly increased relative to *9311*. Increased lignin content may be one of the reasons for the enhanced resistance of *g380*.

Phenylalanine ammonia lyase (PAL) is the first committed step in monolignol biosynthesis and the phenylpropanoid pathway [15]. Apart from cell wall construction and differentiation, plant peroxidases are also implicated in the defense against pathogens [32]. Studies in tobacco and rice have shown that the expression of *PAL* genes can affect the plant–pathogen interaction [33,34], and PAL-suppressed tobacco did not establish systemic acquired resistance [35]. Transcription of *OsPAL1* and *OsPAL4* is much higher in *g380* than in *9311*.

However, both blast and bacterial blight resistance of the diterpenoid *c2BGC* deletion mutant were attenuated before lesion formation [13]. Our study also showed that lignin content was lower in *g380* before the formation of lesions (Appendix A). However, ROS has accumulated before lesion formation. It is possible the defense response has been activated before the appearance of lesions, but not enough to resist the pathogens. At the same time, we also noted that *g380* senescence was slower than that of the WT at the filling stage, although higher ROS accumulated in *g380*, indicating that ROS accumulation did not affect the aging process of *g380*. This means that as a signal molecule, the accumulation level of ROS is very important to induce rice senescence, or other factors additional to ROS are needed for this process. In addition, according to the “source-sink-transportation” theory, we inferred that the generation of *g380* lesion leads to poor nutrient “transportation”, leading to the competition of leaf tissue as a “source” for more nutrients, which results in the greener leaves of *g380*, and the decrease in the seed setting rate of grains as a “sink”.

In conclusion, we speculated that the large fragment deletion of *c2BGC* will lead to the activation and runaway of defense response, which will lead to ROS accumulation, the up-regulation of *PR* gene, cell death and the formation of lesions, the up-regulation of other diterpenoid genes, the increase in lignin biosynthesis, and the enhanced resistance to bacterial blight.

## 4. Materials and Methods

### 4.1. Plant Materials and Growing Conditions

The *g380* mutant was isolated from a *^60^Co* γ-ray mutant library (*Oryza sativa* L. ssp. *Indica* cv. *9311*). The newly screened plants with the lesion phenotype were backcrossed with wild-type *9311* and selfed for multiple generations, and the *g380* plants with stable lesion phenotype and non-segregated progeny traits were selected as mutant parents. The wild-type *9311* was obtained from the seed bank of our laboratory. *G380* was crossed with Nipponbare to obtain an F_2_ mapping population. All plants were grown in a paddy field under natural conditions at the China National Rice Research Institute, Fuyang, China.

### 4.2. Measurement of Agronomic Traits and Chlorophyll Content

Twenty individuals were chosen randomly from the wild-type *9311* and *g380* and were used to measure agronomic traits. Plant height and panicle length were measured and panicle number per plant, the primary branch number per panicle, spikelet fertility, and thousand-grain weight were calculated manually. Hypothesis testing was performed using a Welch’s *t*-test with GraphPad Prism version 7.0.0 (for Windows, GraphPad Software, San Diego, CA, USA, www.graphpad.com, accessed on 21 June 2022).

The chlorophyll content was measured by slightly modifying the method of Arnon [36]. The wild-type *9311* and *g380* leaf samples used for the determination of chlorophyll content were taken from the flag leaves at about 60 DAS; the *g380* flag leaves had lesions at this time. Three biological replicates were sampled, with each biological replicate containing flag leaves from three different individuals. The leaf samples were cut into square pieces with a side length of about 0.2 cm, soaked in 80% acetone solution, protected from light for ten hours, centrifuged, and the supernatant was taken to measure the absorbance at wavelengths of 663 nm and 645 nm. Subsequent chlorophyll content calculations were based on Arnon’s method without modifications. Hypothesis testing was also performed using a Welch’s *t*-test.

### 4.3. Linkage Analysis and Chromosome Walking

The F_2_ mapping population derived from the cross between *g380* and Nipponbare was used for linkage analysis. The 128 SSR (simple sequence repeats) and InDel (insertion-deletion) markers from our laboratory that are polymorphic in the *9311* and Nipponbare genomes were selected. Taking the F_2_ parents (*g380* and Nipponbare) as a reference, linkage marker screening was performed on a mixed pool composed of equal concentrations of gDNA of eight individuals with the lesion phenotype in the F_2_ population. We then genotyped 104 lesion phenotypic individuals in the F_2_ population using screened linkage markers to complete the primary mapping. The 56 additional InDel markers were designed (and given IDs starting with “L”) in the primary mapping interval based on genome sequence polymorphisms between *9311* and Nipponbare. Seven polymorphic primers were screened and used to genotype 2799 lesion phenotypic individuals in the F_2_ population. Later, some of these designed markers were also used for chromosome walking to confirm the exact location of the deleted region in *g380*. Next, we designed PCR primers with an amplicon size of about 1 kb near the markers that we found at both ends of the deletion, which overlap each other to ensure that the endpoints of the deleted region are not missed. These primers were used to check whether any amplicons were generated from *g380*, to finally determine the deletion site. Finally, the deletion area of *g380* was determined by sequencing. The primers used in the linkage analysis and chromosome walking are listed in Appendix A.

### 4.4. DAB Staining and ROS Related Physiological Indexes Measurement

DAB staining was performed to detect hydrogen peroxide accumulation [37]. The flag leaves from *9311* and *g380* were collected around 60 DAS and immersed in 10% sodium dodecyl sulfate (SDS) solution for about 15 min to remove the wax on the surface of the leaves. Then the leaves were rinsed with water and immersed in 1 g/L DAB dye solution for 6–8 h. Afterward, the leaves were boiled in 95% ethanol until they were completely depigmented. Pictures were taken of the leaves with a scanning imager from Hangzhou Wseen Testing Technology Co., Ltd. (Hangzhou, China).

ROS-related physiological indexes, including H_2_O_2_ and MDA contents, and CAT, POD, and T-SOD activities were measured following the instructions of the Nanjing Jiancheng kit (Nanjing Jiancheng Bioengineering Institute, Nanjing, China. Code No. A045-2, A064-1, A007-1, A084-3, A001-3, and A003-1). Samples for measurement were collected from flag leaves of *9311* and *g380* at about 60 DAS. Six replications (three biological replicates with two technical replicates) were performed for each sample of the five indexes. All data collected were analyzed using a Welch’s *t*-test in GraphPad Prism 7.0.0 for a difference significance test.

### 4.5. qRT-PCR Expression Analysis

In this study, most samples for total RNA extraction were collected from flag leaves of *9311* and *g380* plants at about 60 DAS. To detect the expression of diterpenoid biosynthesis genes before *g380* lesion formation, samples were instead taken of newer *g380* and *9311* leaves at about 30 DAS (at this time, no lesion formed on any leaves of *g380* plants).

cDNA was synthesized from total RNA using the ReverTra Ace qPCR RT Master Mix with gDNA Remover (Toyobo, Japan, Code No. FSQ-301) according to the manufacturer’s instructions. TB Green-based qRT-PCR reactions (TB Green^®^ Premix Ex Taq™ II, Takara, Japan) were performed in a LightCycler 480 II (Roche, Sweden). Six replications (three biological replicates with two technical replicates) were performed for each sample of each gene. For relative quantification, gene expression was calculated by the 2^−ΔΔCt^ method. The rice *Actin* gene (*LOC_OS03g50885*) was used as the internal control for the levels of cDNA used. The difference significance test of data was determined by Welch’s *t*-test in GraphPad Prism 7.0.0. The primers used for qRT-PCR are listed in Appendix A.

### 4.6. Disease Evaluation

Three *Xanthomonas oryzae* pv. *Oryzae* (*Xoo*) strains, *Xoo-173*, *Xoo-339*, and *Xoo-347*, were used to evaluate bacterial blight resistance. The *9311* and *g380* plants were grown in the greenhouse at the China National Rice Research Institute, Fuyang, China, and flag leaves were used for inoculation based on the clipping method at around 60 DAS. The lesion length was measured 15 d after inoculation. More than six replications were performed for each sample of the three strains. Welch’s *t*-test in GraphPad Prism 7.0.0 was conducted on the lesion length data to test the significance of the difference.

### 4.7. RNA-Seq and Data Analysis

RNA extraction was undertaken on flag leaf tissues from *9311* and *g380* plants. Total RNA was extracted and purified using Trizol reagent (Invitrogen, Carlsbad, CA, USA) and RNeasy Plant Mini Kit (Qiagen, Valencia, CA, USA), respectively. After extraction, we used the Nanodrop 2000c to quantify the RNA, and RNA quality was then examined using a Bioanalyzer 2100 (Aligent, Santa Clara, CA, USA). First-strand cDNA was generated using reverse transcriptase and random primers. After synthesis of second-strand cDNA and adaptor ligation, 200 bp cDNA fragments were isolated using gel electrophoresis and PCR amplified by 18 cycles. Library sequencing was undertaken with an Illumina HiSeq2000 instrument.

After obtaining the RNA-seq raw data, high-quality clean data were obtained by removing the adaptor sequences, reads with more than 10% unknown bases (N), and low-quality reads containing more than 50% of bases with Q < 30 and trimming low-quality bases (Q < 30) from the 5’ and 3’ ends. The clean reads were then mapped onto the Nipponbare reference genome (Rice Genome Annotation Project) using Tophat (v 2.0.5). The gene expression levels were quantified in terms of FPKM (fragments per kilobase of exon model per million mapped reads). The R Programming Language and package edgeR [38] were used to perform differential expression analysis with FDR < 0.05 and an estimated absolute log_2_(FC) > 1.

### 4.8. GO and KEGG Enrichment Analysis

We used the SEA method of agriGO v2.0 [39] to perform GO (Gene Ontology) enrichment analysis on the obtained up-regulated DEGs. The R Programming Language and clusterProfiler package [40] were used for KEGG (Kyoto Encyclopedia of Genes and Genomes) pathway enrichment analysis. The relevant parameters were organism = ‘dosa’, pAdjustMethod = ‘BH’, qvalueCutoff = 0.05; the rest of the parameters were default. The KEGG pathway map was completed by submitting the data to the KEGG database and using KEGG Mapper.

### 4.9. Lignin and Diterpenoid Content Measurement

Lignin content was determined using a Lignin Content Assay Kit (Catalog Number AKSU010M) purchased from Beijing Boxbio Science & Technology Co., Ltd. (Beijing, China). The samples used for lignin content determination were taken from *9311* and *g380* flag leaves at about 60 DAS. Three biological replicates were taken from *9311* and *g380*. More than three individual plants were mixed for each biological replicate, and each individual could not be selected repeatedly. Sample treatment and content calculation were carried out according to the instructions of the kit. The data were submitted to Welch’s *t*-test in GraphPad Prism 7.0.0 for significant difference analysis.

The samples for determination of diterpenoid content were selected from *9311* and *g380* plants planted in the same field in two stages with a sowing date interval of about 30 days. Four groups of samples were selected: the newest leaves of *g380* without lesions at about 30 DAS and the newest leaves of *9311* in the same period; the flag leaves of *g380* with lesions at about 60 DAS and the flag leaves of *9311* in the same period. Three biological replicates were taken for each group of samples, no less than six individual plants were collected for each biological replicate, and each individual plant was selected without repetition. The leaf samples were quickly frozen in liquid nitrogen after being cut off and then lyophilized with a lyophilizer. The lyophilized leaves were ground with a Mixer Mill MM 400 (Retsch). Subsequent sample extraction and LC-ESI-QQQ-MS/MS system analysis were performed according to the method of Chen et al. [41]. The obtained relative contents of the same diterpenoids were accumulated, and Welch’s *t*-test in GraphPad Prism 7.0.0 was used to verify whether the difference was significant.

## Figures and Tables

**Figure 1 ijms-23-07234-f001:**
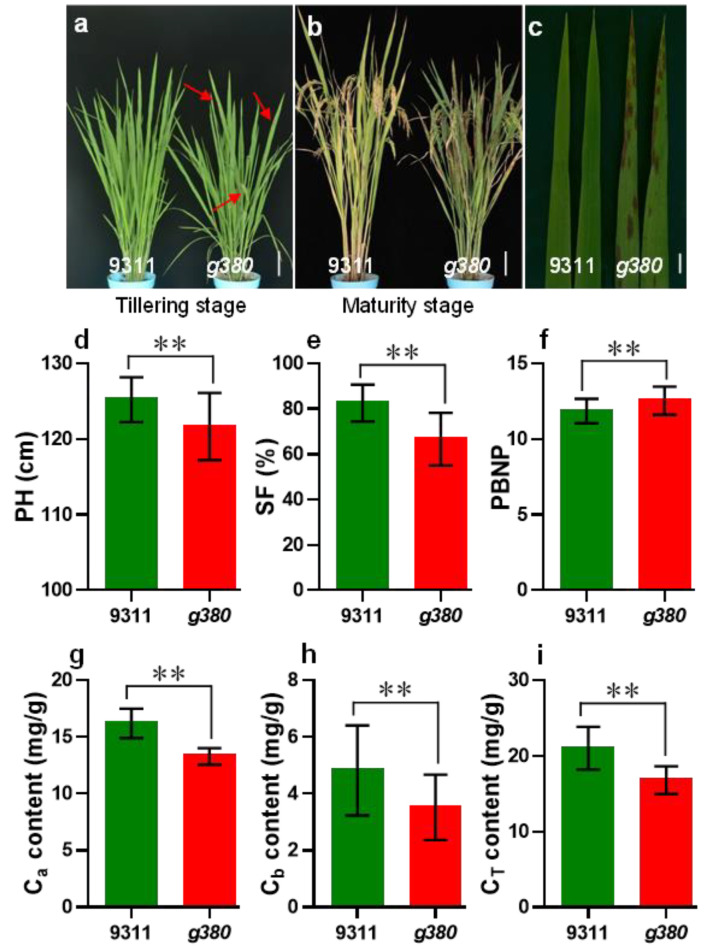
Characterization of the *g380* mutant. (**a**) Plants of *9311* and *g380* at the tillering stage. (**b**) Plants of *9311* and *g380* at the maturity stage. (**c**) Leaves of *9311* and *g380*. (**d**–**f**) Agronomic traits of *9311* and *g380* plants. PH, plant height; SF, spikelet fertility; PBNP, primary branch number per panicle. (**g**–**i**) The chlorophyll content of *9311* and *g380* leaves. Bar = 10 cm in (**a**) and (**b**); red arrows point out lesions on *g380* leaves in (**a**); bar = 1 cm in (**c**). ** represent *p* < 0.01 by Welch’s *t*-test.

**Figure 2 ijms-23-07234-f002:**
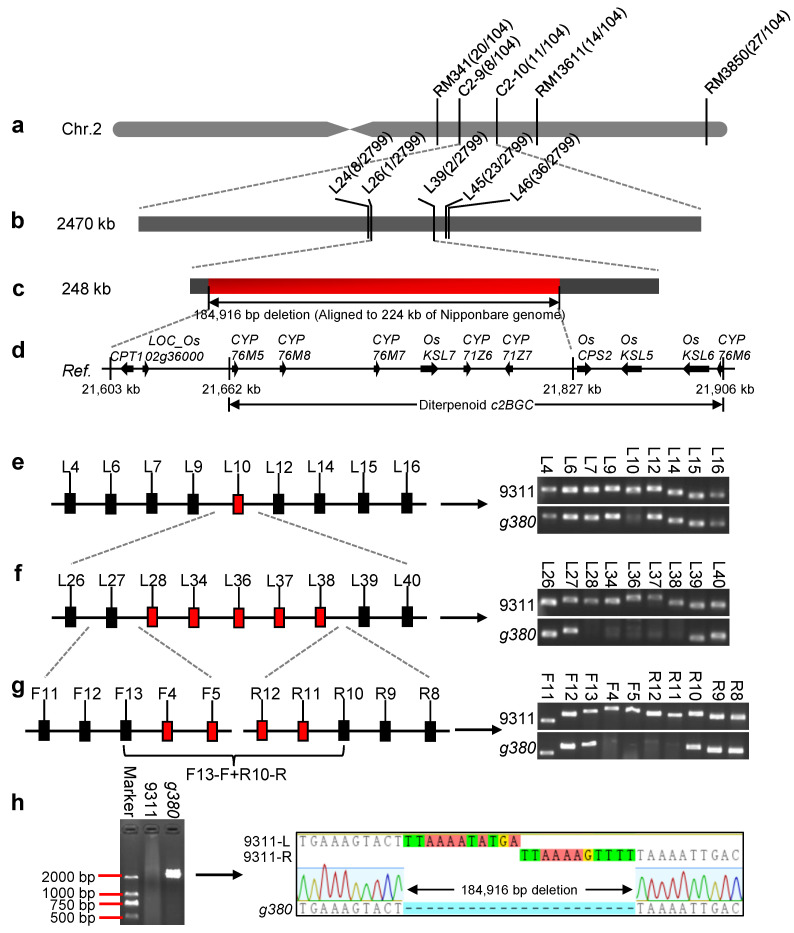
Fine mapping of *g380*. (**a**,**b**) Preliminary mapping interval. (**c**) Fine mapping interval; red area represents the deletion fragment. (**d**) Reference genes in the deletion area. (**e**–**g**) Process to determine the deletion area in *g380*; general view of the process is on the left, and the results of agarose gel electrophoresis are on the right. (**h**) Results of *9311* and *g380* amplified with F13 forward primer and R10 reverse primer; the start and end sequences of the deleted fragments were obtained by sequencing. Red boxes in (**e**–**g**) represent the deletion area or primers in the deletion area.

**Figure 3 ijms-23-07234-f003:**
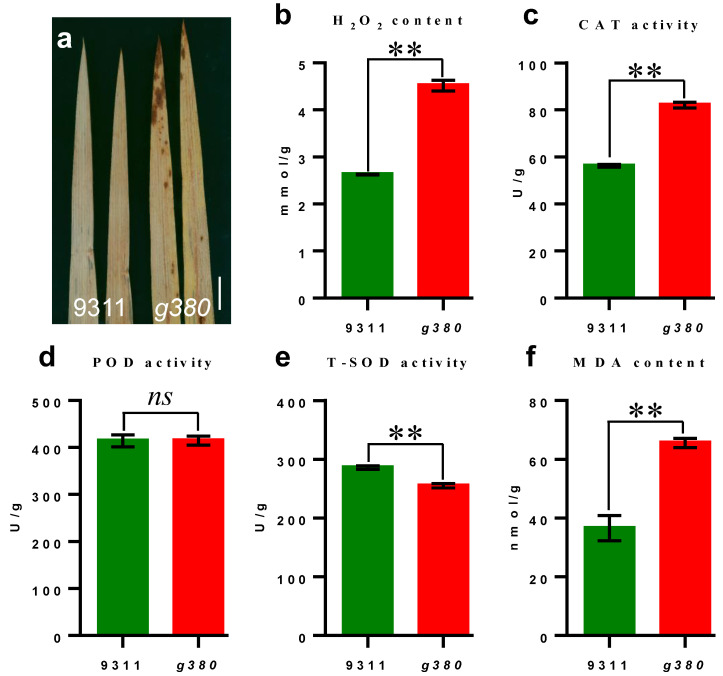
ROS accumulation in *g380* after the lesion formation. (**a**) DAB staining of *9311* and *g380* leaves. (**b**–**f**) The ROS related physiological–biochemical indexes of *9311* and *g380* leaves. Bar = 1 cm in (**a**). Values are mean ± SD (*n* = 6). ** indicates significant differences at *p* < 0.01 level by Welch’s *t*-test.

**Figure 4 ijms-23-07234-f004:**
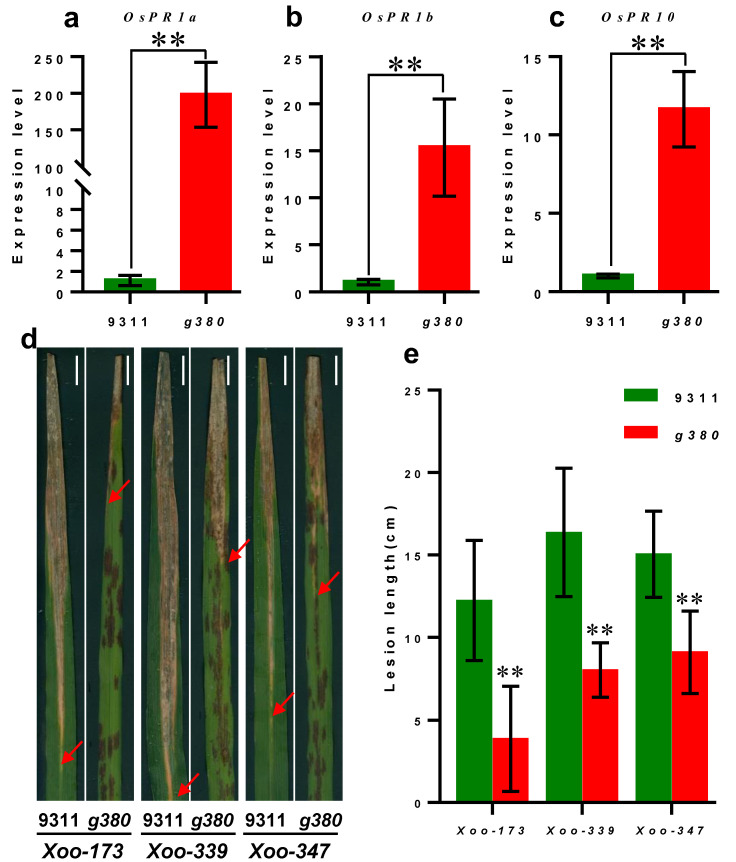
The expression levels of *PR* genes and the result of bacterial blight inoculation. (**a**–**c**) The expression levels of *PR* genes; values are mean ± SD (*n* ≥ 6). ** indicates significant differences at *p* < 0.01 level by Welch’s *t*-test. (**d**) Phenotypes of representative leaves from the *9311* and *g380* plants at 15 d after infection with *Xoo-173*, *Xoo-339*, and *Xoo-347*; bar = 1 cm. (**e**) Comparison of the lesion lengths on leaves from *9311* and *g380* plants at 15 d after infection with *Xoo-173*, *Xoo-339*, and *Xoo-347* (*n* ≥ 6); ** means *p* < 0.01 level by Welch’s *t*-test.

**Figure 5 ijms-23-07234-f005:**
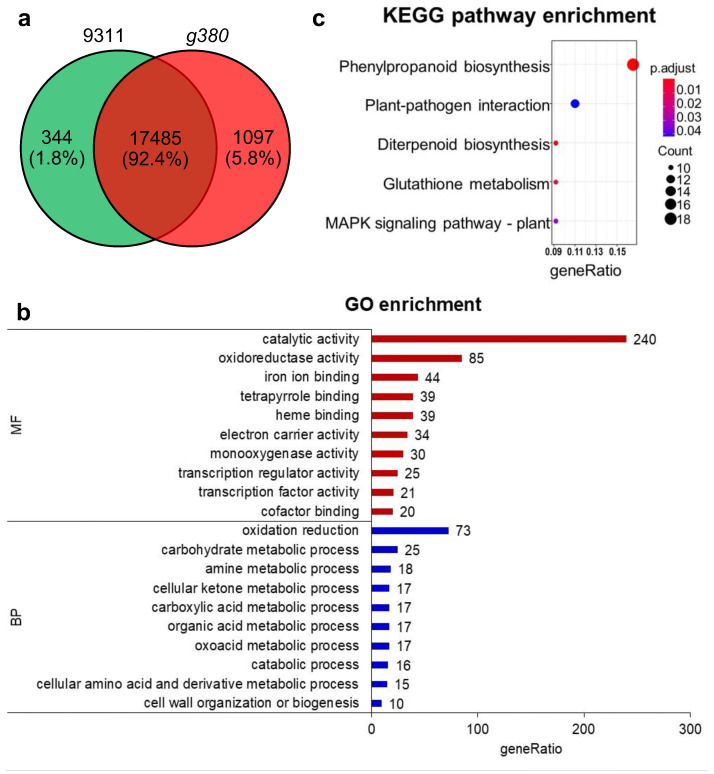
Expressed gene statistics and GO and KEGG pathway enrichment analysis of DEGs. (**a**) Venn diagram of *9311* and *g380* expressed genes. (**b**) GO enrichment analysis of upregulated DEGs; the top ten terms enriched in BP and MF. BP: biological process; MF: molecular function. (**c**) KEGG pathway enrichment analysis of up-regulated DEGs.

**Figure 6 ijms-23-07234-f006:**
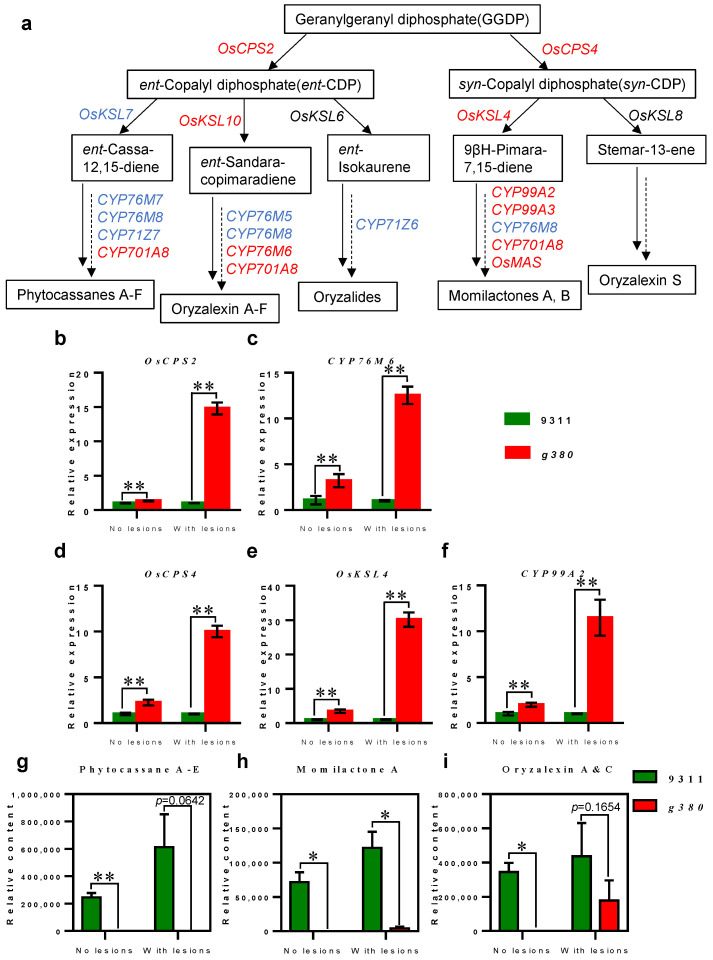
Decreased diterpenoid biosynthesis in *g380* leaves. (**a**) Biosynthesis process of major diterpenoid phytoalexins in rice; substances in black bordered text boxes are metabolites; genes indicated in blue are deleted in *g380*, genes indicated in red are DEGs upregulated in *g380*, genes indicated in black have no significant difference in expression level between *9311* and *g380*. The solid arrows represent the direct response, and the “solid + dashed” arrows represent that the process has not been explained sufficiently clearly. (**b**–**f**) Expression of non-deleted diterpenoid biosynthetic genes in *9311* and *g380* before and after the formation of *g380* lesions; (**b**,**c**) display genes located on chromosome 2; (**d**–**f**) display genes located on chromosome 4; error bar represents ± SD; ** indicates *p* < 0.01 by Welch’s *t*-test. (**g**–**i**) Diterpenoid phytoalexin content in *9311* and *g380* leaves before and after *g380* lesion formation; error bar represents ± SE; * indicates *p* < 0.05, ** indicates *p* < 0.01 by Welch’s *t*-test.

**Figure 7 ijms-23-07234-f007:**
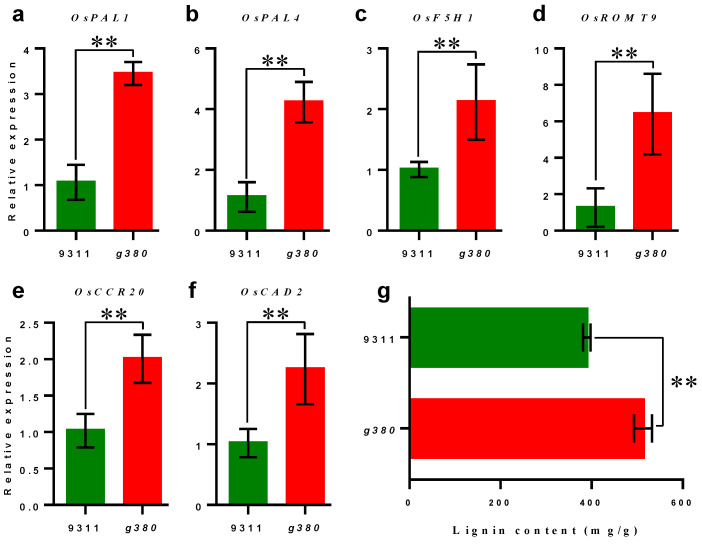
Increased lignin biosynthesis in *g380*. (**a**–**f**) The expression levels of lignin biosynthetic genes in *9311* and *g380* leaves. (**g**) Lignin content in *9311* and *g380* flag leaves. All data presented as mean value ± SD; hypothesis testing was performed with Welch’s *t*-test, *n* = 6, ** means *p* < 0.01.

## Data Availability

Not applicable.

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
