# Peer review of "Deletion of Diterpenoid Biosynthetic Genes CYP76M7 and CYP76M8 Induces Cell Death and Enhances Bacterial Blight Resistance in Indica Rice ‘9311’"

_ijms, 2022, doi:10.3390/ijms23137234_

Round 1

Reviewer 1 Report

Authors suggest that a cluster of genes responsible for the biosynthesis of diterpenoids is located on chromosome 2, and that this cluster is involved in immunological reprogramming, which in turn affects cell death in rice. Overall, the manuscript is well written and it provides valuable insights, therefore it can be accepted for publication in the present after a careful English language and spelling check.

Reviewer 2 Report

The authors identified the causal mutation associated with lesion mimic mutants (LMMs), g380 through chromosome walking mapping method. Deletion of 184,916 bp containing 8 genes on chromosome 2 were founded in g380 mutant and, out of 8 genes,  two genes CYP76M7&CYP76M8 involved in diterpenoid biosynthesis were identified as the causal genes for LMM with support of the recent publication data (Li et al. 2022). Although many research results are overlapped with the previous publications (Li et al. 2022; Ye et al. 2018), the authors presented further insight of the LMMs such as transcriptome analysis by using RNA-seq and expression validation by using qRT-PCRs. However, the manuscript has to be re-checked and improved before publication.

·         There are two Figure 1s. (Characterization of the g380 mutant, Fine mapping of g380). It has to be revised. Following with this, all the Figure numbers in the main text must be checked. In addition, numbering of all the supplementary data have to be checked and revised.

·         In ‘Linkage analysis and chromosome walking’ section (Line 373), the authors have to describe the number of SSR markers screened and the number of polymorphic SSR markers between the mutant and Nipponbare for bulked segregation analysis.

·         Figure 2h: To provide solid data, chromatogram of the Sanger sequencing results at the junction points (near deletion region) should be compared between 9311 and g380 mutant as a figure and it should be presented in Figure 2h or in the supplementary data.

·         Full names of the abbreviations including MDA, CAT, POD, and T-SOD should be shown at where it is shown first time in the main text. But, the full names of these were given later in the Material and Methods sections. It should be revised.

·         Species names (Line 2 and line 99), gene names, and mutant names should be italic but it is mixed in the current version of manuscript. It has to be checked throughout the manuscript.

Reviewer 3 Report

Authors screened a lesion mimic mutant, which exhibited enhanced ROS and lignin accumulation and demonstrated resistance against Xoo. My main concern is the novelty of the work. Since the deletion of CYP76M7 and M8 have already been shown to result in lesion phenotype and associated with bacterial resistance, I do not find any other selling point of this MS. This work can however be made interesting by considering my following comments:

What is the biochemical/physiological factor inducing lesion in g380 mutant? If it is ROS, could you compare the lesion phenotype with other ROS overproducing mutants?

What is the reason for the onset of lesions at 40 DAS and not at early plant stages? Please comment and discuss. If the lesions develop late at tillering stage, 40 days after sowing, does the appearance of lesions also affect bacterial resistance?

Since ROS here has been postulated as the main mechanism of defense and causal agent for the appearance of lesions, the real-time change in ROS dynamics with plant age needs to be demonstrated  The same comment goes for trend for lignin deposition across different stages, with and without infection.

Since ROS has also been postulated as a key driver of plant growth (cell division, cambium and vasculature control), it would like to see the real-time changes in ROS production in g380 over WT and associate the change, if any with differential plant growth.

Since authors show enhanced/increased PAL flux with enhanced PR, WRKY and MAPK signaling genes, it is worthwhile to demonstrate the changes in salicylic acid accumulation.

The accumulation pattern of diterpenoids is not consistent with the OSCPS4 branch of synthesis, since all the genes are upregulated there (Figure 5 d, e, f). please provide a pertinent explanation and discuss.

Minor issues:

ROS accumulation at the reported levels is seemingly detrimental to plant growth. But in Figure 1 b I see the plant looks greener (healthy) than the WT (off course barring the height difference). Please comment and include in the discussion.

The greener plant phenotype for g380 in Figure 1b could be affecting the plant senescence and resource remobilization. Please discuss this aspect as well.

Please check the citation for Figure 1. It is confusing as to which one is your actual figure 1.

For figure 1, please mention the plant age/stage at which the agronomic traits were assessed

Round 2

Reviewer 2 Report

The manuscript was accordingly revised and improved in the revised version.